# Association of Hypertension and Organ-Specific Cancer: A Meta-Analysis

**DOI:** 10.3390/healthcare10061074

**Published:** 2022-06-09

**Authors:** Morgan Connaughton, Mahsa Dabagh

**Affiliations:** Department of Biomedical Engineering, University of Wisconsin, Milwaukee, WI 53211, USA; connaug3@uwm.edu

**Keywords:** cancer, hypertension, comorbidity, MMP, statistical analysis

## Abstract

Hypertension and cancer are two of the leading global causes of death. Hypertension, known as chronic high blood pressure, affects approximately 45% of the American population and is a growing condition in other parts of the world, particularly in Asia and Europe. On the other hand, cancer resulted in approximately 10 million deaths in 2020 worldwide. Several studies indicate a coexistence of these two conditions, specifically that hypertension, independently, is associated with an increased risk of cancer. In the present study, we conducted a meta-analysis initially to reveal the prevalence of hypertension and cancer comorbidity and then to assess which organ-specific cancers were associated with hypertension by calculating the summary relative risks (RRs) and 95% confidence intervals (CIs). Our analysis shows that hypertension plays a role in cancer initiation. Our extended analysis on how the hypertension-associated angiogenesis factors are linked to cancer demonstrated that matrix metalloproteinases 2 and 9 appear to be two key factors facilitating cancer in hypertensive patients. This work serves as an important step in the current assessment of hypertension-promoted increased risk of 19 different cancers, particularly kidney, renal cell carcinoma, breast, colorectal, endometrial, and bladder. These findings provide new insight into how to treat and prevent cancer in hypertensive patients.

## 1. Introduction

Hypertension, also known as high blood pressure, is a complex chronic condition that affects approximately 45% of the United States population [1] and is a growing condition globally, particularly in European and Asian countries [2]. Hypertension contributes to many different cardiovascular diseases and complications due to the presence of hypertension can lead to chest pain, irregular heartbeat, heart attack, or heart failure. Hypertension can also cause a stroke by blocking the oxygen supply to the brain and also cause kidney disease, eventually causing kidney failure [3]. On the other hand, cancer, as one of the leading causes of death worldwide, has accounted for almost ten million deaths in 2020, and the death rate from cancer is projected to continue to rise [4]. While both cancer and hypertension are expected to rise, it is important to understand the probability of comorbidity between these chronic conditions. More specifically, is there an association between hypertension and an increased risk of incidental cancer at organ-specific sites? In this study, we sought to answer questions regarding organ specificity, gender, and ethnicity.

A previous meta-analysis showed an association between hypertension and kidney cancer, colorectal, liver, endometrial, esophageal adenocarcinoma, and a borderline significance of breast cancer [5]. The studies in [5] included 148 studies up until 2017 and looked at both studies that had no association and studies that had a positive association. The impact of hypertension on breast, colorectal, and endometrial cancers has also been reported in other studies [6,7,8,9,10,11,12,13,14,15,16,17,18,19,20,21,22,23,24], but the most confirmed association has been made for renal cell carcinoma (RCC) [5,25,26]. One study observed the odds ratio (OR) with a 95% confidence ratio in individuals with self-reported hypertension in both men and women and found that hypertension was independently associated with a 40% increased risk of RCC compared to the refenced nonhypertensive patients (OR: 1.4; CI: 1.1–1.9 and OR: 1.0; CI: reference, respectfully) [26], and this increase was also observed in younger (20–39 years of age) hypertensive patients, where they saw a smaller but significant 16% increase in risk [27]. Another study assessed the role of hypertension in the race disparity of RCC incidence and found that if causal, removing the exposure (i.e., hypertension) would decrease the RCC incident rate by 44% and 35% for African American men and White men, respectively, and 51% and 30% for African American and White women, respectively [28]. Hypertension has been identified as a sole statistical risk factor out of the other metabolic risk factors for kidney cancer pathology including increased tumor size, higher tumor grade, and trends towards a higher tumor stage [29].

After further analysis, previous studies have shown that not only does hypertension possibly promote incidental cancer risk, but prehypertension (SBP: 120–139 mmHg or DBP: 80–89 mmHg) and antihypertension treatment may also promote increased risk of incidental cancer [29,30]. One study showed that prehypertension and hypertension were both associated with developing esophageal cancer [30]. Another study saw an association of prehypertension and increased cancer risk only in men for all cancers, but the strongest association was with colorectal cancer [16]. Since previous studies have demonstrated that hypertension may play a role in cancer, the present study purely focused on untreated hypertensive patients to better understand the pathology between the two and examine possible crosslinks between these highly complex diseases to emphasize the importance of preventative strategies.

Carcinogenesis, or the initiation of cancer, is understood, but a universal theory as to why it occurs is unclear. One possible theory may be associated with hypertension. In addition to cancer initiation, previous studies have also shown that hypertension may also play a role in cancer progression as well. In a study involving bladder cancer, high blood pressure was associated with increased risk and mortality among men, and blood pressure showed a graded increase in strength of association from overall bladder cancer, NMIBC, MNIBC, and bladder cancer mortality [31]. Dose-dependent SBP has been associated with bladder cancer mortality in an analysis of never-smokers, which is important to note, as smoking is one of the largest risk factors for bladder cancer (accounting for approximately 50% of cases) [32]. Other studies have also concluded that hypertension induces a poor overall survival rate and an increased risk of cancer mortality [6,31,32,33,34,35,36,37,38,39,40,41], for cancers of the oropharynx, rectum, pancreas, lung, prostate, bladder, and kidney for men and pancreas, breast (women with the highest combined blood pressure tertial were at a 41% increased risk of total mortality compared with women in the lowest tertial [39]), endometrial, and malignant melanoma for women and esophagus cancers for both [6].

With newer studies being introduced relating to the development of cancer in hypertensive patients, the aim of this study was to conduct a meta-analysis of the data up to the present (June 2021) to evaluate the association of untreated hypertension and increased risk of developing patient-specific, organ-specific cancer and the possible pathologies behind the association.

## 2. Methods

### Literature Review and Search Criteria

Previous works have used machine learning on cancer and hypertension as keywords; however, for this study, we wanted to include only studies showing hypertension and an increase in incidental cancer risk. Therefore, a manual search was conducted to find data, which took five months. To conduct the literature review needed for this study, databases, such as PubMed and Embase, were used to identify studies published up until June 2021. The study used the following search criteria: (hypertension OR high blood pressure OR metabolic syndrome) AND (cancer OR incidental risks OR neoplasms OR organ-specific cancer sites: breast, RCC, endometrial, GI, prostate, and colon). This search process was used to identify all qualifying studies that involved clinical cohort and/or case-controlled studies that considered the systolic blood pressure (SBP) and diastolic blood pressure (DBP) and their relation to any organ-specific cancer sites. After the initial search that took place from January 2021 until the end of March 2021, the search was then specified for articles from 2017 until the present to find new data on the topic, since the previous meta-analysis included data up until 2017. This manual search was concluded in June 2021.

If the study was on metabolic syndrome, which is a combination of factors, such as BMI, high triglycerides, and hypertension, only hypertension data were examined and extracted. Hypertension was defined as being higher than the baseline blood pressure given in each individual study, and the data were given either by measurement, medical history, or from a questionnaire. The study criteria included the following guidelines: (1) case-control or cohort study design; (2) the study reported either a relative risk, hazard ratio, or odds ratio that could be converted to a relative risk; (3) the exposure was hypertension, and the outcome was incidental cancer risk.

## 3. Data Extraction

The data were extracted manually by the reviewers, who first read the abstract and title of the study. The author’s name, location of study, year of publication, design of study, and the definition of hypertension were all independently extracted. In total, there were 30 cohort studies, 16 case-control studies, and 1 unknown.

## 4. Statistical Analysis

The data from previous studies on the individual cases were provided by statistical software and either gave relative risks or an odds ratio, depending on what the individual study analyzed. Primary analysis calculated summary relative risks and confidence intervals of 95%. A few studies that were included used odds ratios, but in this case, they were then converted to relative risks [42,43]. A subgroup analysis was also conducted of dose-dependent SBP or DBP per 10 mmHg and hypertension. The present study also evaluated the statistical heterogeneity among studies using the Cochran’s Q test and *I*^2^ test for studies with two or more exposure and outcomes [44,45]. The statistical analysis was conducted using Microsoft Excel [46].

## 5. Results

### 5.1. Characteristics

Figure 1 displays a flowchart of eligible studies depicted in the meta-analysis. Initially, 73,652 citations were screened after searching databases with the keywords. No automation tool was used throughout the entire screening. A total of 80 studies were initially identified during the time period of the manual literature search from January 2021–March 2021. The main criteria of the study were to include only clinical work (i.e., no reviews or meta-analysis) that showed a positive association between hypertension and an increased risk of incidental cancer risk and to focus on works published after 2017. Based on our criteria, of these 80 studies, 5 were removed because they were duplicates (n = 75), 30 were removed for not meeting abstract criteria (n = 45), for 3 we were unable to retrieve full access (n = 42), and 17 were excluded for discussing pathology, mortality, previous meta-analysis/review, having no blood pressure information, and including antihypertension drugs. This method yielded 25 included studies. Since the previous meta-analysis [5] included 148 studies only up until 2017, another hand search was conducted to specifically identify studies from 2017 to June 2021. After conducting a hand search specifically for studies from 2017 to 2021, 22 studies were identified. In total 47 studies were identified without the use of machine learning [6,7,8,9,10,11,12,13,14,15,16,17,18,19,20,21,22,23,24,25,26,27,28,30,31,32,47,48,49,50,51,52,53,54,55,56,57,58,59,60,61,62,63,64,65,66,67]. Of these 47 studies, 22 were published after 2017, as one of our focuses was to obtain more recent publications as new data have been introduced over the past 5 years since the previous meta-analysis was conducted [5].

The included studies ranged from 1998 to 2021. In these studies, there were 678,949 patients and 35,499,496 participants. Hypertension was then defined in 41 of the studies (87%) using WHO criteria (BP ≥ 140/90 mmHg; n = 12; 29%), NCEP-ATPIII criteria (BP ≥ 130/85 mmHg; n = 7; 17%); 13 (32%) used self-reporting data; 9 (22%) used other criteria. To obtain the exposure (hypertension) assessment in the studies, blood pressure was either measured (n = 22; 49%), administered through a questionnaire (n = 14; 31%), or obtained from medical records (n = 9; 20%). Only 45 (96%) of the 47 studies included these measurements. The studies were located in Europe (n = 18; 38.3%), North America (n = 10; 21.3%), Asia (n = 17; 36.2%), and South America (n = 2; 4.2%).

### 5.2. Overall Analysis

The overall analysis for the meta-analysis included 47 studies, and the combined overall effect estimate (RRs) and heterogeneity of these studies are presented using a relative risk model for each individual cancer. In total, 19 different cancers were analyzed for the overall meta-analysis, as seen in Table 1 and Table 2. The subgroup analysis that depicted the dose-dependent manner of SBP and DBP contained eight cancers, but only two cancer types (i.e., muscle-invasive bladder cancer and RCC) contained more than one study; therefore, heterogeneity was not calculated for the remaining six.

### 5.3. Evidence

There were associations made between almost every cancer site evaluated, but when accounting for the number of studies involved (e.g., lung: n = 1), those with only one study were not included in our results as a positive association. When looking at the data with numerous studies involved, we found statistically significant associations between hypertension and risk of incidental kidney cancer for both men (Figure 2; n = 4 prospective studies; summary relative risk (RR): 2.05; 95% CI: 1.50–2.90; *I*^2^: 94.4%) [6,7,22,27] and women (Figure 2; n = 2; RR: 1.83; 95% CI: 1.44–2.33; *I*^2^: 58.1%) [10,22]. We observed a large heterogeneity in kidney cancer for men (*I*^2^: 94.4%) and a smaller between-study heterogeneity for kidney cancer in women (*I*^2^: 58.1%). RCC also saw an increase in risk (Figure 2; n = 3; RR: 1.60; CI: 1.48–2.58; *I*^2^: 57.3%). [26,28,64] in men and a weak association (Figure 2; n = 2; RR: 1.31; CI: 0.95–2.15; *I*^2^: 0%) [26,64] in women. When comparing these data with a previous meta-analysis on RCC, similar results were obtained [68].

Breast cancer also showed a positive association between hypertension and cancer risk (Figure 2; n = 9; RR: 1.65; CI: 1.30–3.55; *I*^2^: 69.1%) [9,11,12,13,14,15,20,21,23]. This organ-specific site contained the most outcome estimates, and the between-study heterogeneity was higher (*I*^2^: 69.1%). One study was not included because it did not include data on the confidence intervals, but it did observe an odds ratio of 2.5 for women with hypertension developing breast cancer [63]. There was also a statistically positive association between hypertension and colorectal cancer (Figure 2; n = 6; RR: 1.62; CI: 1.13–1.77; *I*^2^: 64.3%) [6,16,17,18,19,22] and endometrial cancer (Figure 2; n = 5; RR: 1.41; CI: 1.07–1.89; *I*^2^: 73.2%) [6,7,8,10,24]. Furthermore, we observed that there was a positive association between hypertension and an increased risk of incidental prostate cancer (Figure 2; n = 4; RR: 1.38; CI: 1.20–2.15; *I*^2^: 0%) [20,22,48,51]. Bladder cancer had a weaker positive association (Figure 2; n = 3; RR: 1.21; CI: 0.97–1.50; *I*^2^: 0%) [6,52,56].

In the subgroup analysis of SBP and DBP per 10 mmHg, SBP showed a positive dose-dependent association with RCC (Figure 3; n = 2; RR: 2.26; CI: 1.50–3.62; *I*^2^: 92.9%) [25,65] and muscle-invasive bladder cancer (MIBC) (Figure 3; n = 2; RR: 1.29; CI: 1.05–1.57; *I*^2^: 0%) [31,32]. DBP also showed a positive dose-dependent association with RCC (Figure 4; n = 2; RR: 2.28; CI: 1.50–3.65; *I*^2^: 90.8%) [25,65].

## 6. Discussion

This meta-analysis observed hypertension and its association with increased cancer risk in 19 different cancers. A strong positive association was observed for the following: kidney, RCC, breast, colorectal, endometrial, and bladder. There also was a positive association of SBP per 10 mmHg for MIBC and DBP per 10 mmHg. With the other cancers only containing one study each, a valid and generalized association could not be made.

Several other studies have investigated this association and have shown positive results, especially hypertension and the risk of RCC [26,28,64,69,70,71,72,73], but cancers such as breast cancer have been controversial. Some studies either observed no association with hypertension and breast cancer risk or the association was only with hypertensive patients on antihypertension medication [74,75,76,77]. Specifically, there was a previous study that found that there was an increased risk of developing ER+ breast cancer with those who were long-term (≥5 years) users of antihypertension drugs (RR = 1.21; 95% CI: 1.03–1.43) [75]. A recent meta-analysis showed a positive association, and that hypertension increased the risk of incidental breast cancer by 15% [78]. Moreover, we found that even when adjusting for menopausal status, the association increased for postmenopausal women [44]. Through our further analysis [9,11,12,13,14,15,20,21,23], we saw a significant increase in risk for postmenopausal women, possibly because the present study mostly observed postmenopausal women and only included one study that observed a significant association found between hypertension and breast cancer over the entire sample regardless of menopausal status [12]. To understand the mechanisms underlying hypertension-caused organ-specific cancer, we here discuss the most vital mechanisms.

## 7. Remodeling of Extracellular Matrix

The extracellular matrix (ECM) is a type of noncellular structure that regulates most of the cellular functions of tissues and organs. It impacts cell behavior (i.e., cell proliferation, polarity, differentiation, migration, and adhesion). More specifically, the microvascular ECM structure is essential for tissue repair and wound healing [79]. On the other hand, the ECM is susceptible to remodeling that can be caused by pathologic conditions such as inflammatory diseases (hypertension is associated with inflammation), tissue fibrosis, and cancer. When ECM remodeling occurs, a broad range of changes can occur, including ECM stiffening. Hypertension-induced arterial wall remodeling has been associated with ECM stiffening [80], and ECM stiffness has been shown to induce a malignant phenotype that will disrupt adhesion molecules’ cell-to-cell junctions and increase tumor growth [81]. Looking into identifying the mechanisms that are associated with cancer progression in hypertensive patients and understanding how these mechanisms change the ECM may provide possible explanations for why hypertension patients are more prone to specific types of cancer.

## 8. Vascular Endothelial Growth Factor Regulation

A large-scale retrospective study looked at the overall survival outcome of patients with hypertension and the prognosis of nasopharyngeal carcinomas (NPC). Not only did the results show that those patients with hypertension stage II had a worse overall survival rate than those without, but vascular endothelial growth factor (VEGF) expression also increased in patients with advanced NPC [41]. VEGF is a multifunctional glycoprotein and is an important regulator of physiological or pathological angiogenesis (the process of new blood vessels growing from preexisting vessels, which is important for malignant tumor growth) by increasing blood vessel permeability, proliferation, endothelial cell growth, migration, and differentiation [41]. VEGF is shown to be increased in hypertensive patients and has been linked to tumor progression and poor prognosis in many tumor types [41].

## 9. Reactive Oxygen Species and Renin–Angiotensin–Aldosterone System Regulation

Metabolic syndrome components possibly promote cancer by generating reactive oxygen species (ROS), increasing hormone production and availability, which includes estrogen, insulin-like growth factor (IGF-1), and adipokines. On the other hand, when looking at the individual components of hypertension, it has been related to insulin resistance and, hence, to IGF-1, which is related to cell growth and neoplastic progression [59]. The renin–angiotensin–aldosterone system (RAAS) may also play an important role in RCC risk specifically [82,83]. RAAS is a hormonal mechanism that regulates blood pressure and is closely linked to hypertension. A recent study on RAAS showed that two single-nucleotide polymorphisms (SNPs) of angiotensin II receptor have been associated with RCC [82]. This pathway could explain the associations between hypertension and RCC risk, and possibly also for prostate and breast cancer [9,34].

## 10. Matrix Metalloproteinases Regulation

Matrix metalloproteinases (MMPs) play a role in both hypertension and cancer [84,85,86]. MMPs are enzymes that are capable of degrading collagen in the ECM but now are being looked at for their role in cancer, because alterations of the ECM from both stiffness and degradation contribute to tumor growth and progression [84,85,86]. There are 23 MMPs expressed in humans, and they have been recently shown to be associated with receptor cleavage of the insulin receptor, leading to insulin resistance and, as mentioned above, insulin resistance is typically associated with hypertension [59,87,88]. Two of the most studied, MMP-2 and MMP-9, belong to the MMP subtype gelatinases [86]. MMP-2 and MMP-9 both have pro- and anti-inflammatory effects and, specifically, MMP-9 has been associated with increased arterial stiffness and elevated blood pressure in hypertension. One study showed an approximately 1.97-fold increase in blood pressure progression in individuals with detectable MMP-9 levels [88,89]. MMP-2 also contributes to hypertension-induced arterial wall changes and sustains hypertension [90].

The mechanisms involved in the activation of these MMPs is unclear, but hypertension may contribute to MMP-2 activation [90]. This phenomenon is potentially caused by the vascular remodeling that is induced by mechanical stress that activates the platelet-derived growth factor mechanoreceptor, PDGF-R, and protein kinase signaling pathways [90]. MMP-2 might also be responsible for the adhesion molecule, cadherin’s disruption [90]. Activation of MMPs may also be caused by another type of MMP. For example, MMP-9 is activated by MMP-2-3-13-17 and -26 [91]. This indirect activation of pro-MMP-9 is through the activation of MMP-2 and -13 on the cell surface by membrane type-1 MMP [91]. Another study observed the latent form of pro-MMP-9, when binding to a receptor complex containing a tissue inhibitor of metalloproteinase-1 (TIMP-1) and A disintegrin and metalloproteinase domain-containing protein 10 (ADAM10), is activated by the membrane type-1 MMP/MMP-2 axis, thus stimulating metastasis [92].

MMPs have been shown to also be upregulated in many cancer types [86]. One study aimed to identify patterns of MMP dysregulation and to associate MMP expression to a patient’s survival in all 15 cancer types. That study found that MMP-9 was significantly upregulated in twelve of the fifteen cancer types [86]. Another study showed that MMP-9 expression in breast cancer was a predictor of shorter survival of patients and that MMP-9 was associated with higher tumor grade and a confirmed positive association between MMP-9 and ECM remodeling [93]. MMP-2 and MMP-9 were also upregulated in breast cancer tissue but were not in adjacent normal tissue [94]. Based on our analysis, MMP-9 has the potential to be a significant biomarker for cancer in hypertensive patients.

MMPs’ contribution to hypertension and its association for increased risk of incidental cancer is still unclear, as most studies only observe one condition (either cancer or hypertension), not both simultaneously. Therefore, a clear link cannot be concluded on how hypertension promotes patient-specific organ-specific cancers and what mechanisms drive the comorbidity of these two diseases. Our summary theory, based on an extensive review of over 102 published studies in this field, is that high blood pressure activates MMP-2, where MMP-2 has been shown to activate pro-MMP-9, and MMP-9 has been reported to increase the risk of hypertension. Hypertension-induced increased stiffness and degradation of the ECM has been widely reported. On the other hand, ECM stiffening and degradation are linked to progression of several cancer types. Understanding how hypertension and the MMPs (specifically MMP-2 and -9) affect one another would be beneficial for designing cancer prevention strategies for hypertensive patients.

The present meta-analysis used a wide search criteria that were specifically for studies with a positive association between hypertension and an increased risk of developing incidental cancer. The present study also did not use machine learning, because the researchers wanted to gain more in-depth information initially on the papers for their own gain and knowledge on the topics of both hypertension and cancer. Cancer and hypertension are both extremely complex and may be promoted from various cofounding parameters such as BMI, genetics, diabetes, hypertension, and high triglycerides. A previous study that did look at other factors associated with metabolic syndrome found that the strongest association with prostate cancer was a combination of obesity and hypertension, but when they studied these factors individually, hypertension, independently, was associated the most strongly with prostate cancer risk [48]. Another study evaluated whether the combination of comorbid conditions was associated with epithelial ovarian cancer (EOC), and they found that women who had untreated hypertension for less than 10 years were at the strongest risk. Importantly, they also found that women diagnosed with all three—hypertension, hyperlipidemia, and diabetes—had a decreased risk of EOC [54]. Other studies have also demonstrated that out of all the metabolic factors associated with cancer, hypertension plays a very strong role [17,29,35,36] in cancer initiation. Since previous studies have demonstrated that hypertension may be associated with cancer at a higher risk than other metabolic factors; the present study purely focused on untreated hypertensive patients to better understand the pathology between the two and examine possible crosslinks between these highly complex diseases to emphasize the importance of preventative strategies. Our study serves as an important first step in understanding the association between hypertension and different organ-specific cancer incidents. Both cancer and hypertension, as mentioned, are partly determined by genetic factors. However, identifying any common genetic link between hypertension and cancer is beyond the scope of this study.

The present study did have limitations. It did not look at contributing factors, such as BMI or genetics, in order to gain information purely on hypertension and cancer; therefore, future studies on this research would be beneficial to observe if these contributing factors, along with hypertension, also had an association with developing or not developing incidental cancer. A majority of the studies performed multivariable adjustments, but not all, and the present study did not perform a subgroup analysis. Another possible limitation is the lack of data on the details of each malignant tumor that developed within these patients and the studies lacked information on how long these tumors developed after the initial hypertension examination. Lastly, some organ-specific cancer sites only had one included study; thus, a statistical analysis could not be made. Future research for those organ-specific cancer sites would be beneficial for this work.

In conclusion, since hypertension and cancer are both extremely complex, there are several possible parameters that could also play a role in cancer initiation and progression. The present study focused on hypertension because of its pathological complexity and the lack of information on the possible crosslink between hypertension and cancer. The present study was also able to demonstrate that one of these factors (i.e., hypertension) may contribute to carcinogenesis and that this association between hypertension and cancer is possibly observed between different organ-specific cancer sites. Specifically, the present study observed that individuals with hypertension were at a high risk of kidney, breast, colorectal, endometrial, and bladder cancers. It is important to note that the high complexity of hypertension and cancer makes it difficult to provide a definite remark as to why and how this is occurs, but this study was also able to give detailed evaluations of the possible mechanisms of this observed cross-linkage and attempted to enhance the understanding of this phenomenon and provide detailed information on relevant information that may have the potential to improve preventable and therapeutic treatments for these patients.

## Figures and Tables

**Figure 1 healthcare-10-01074-f001:**
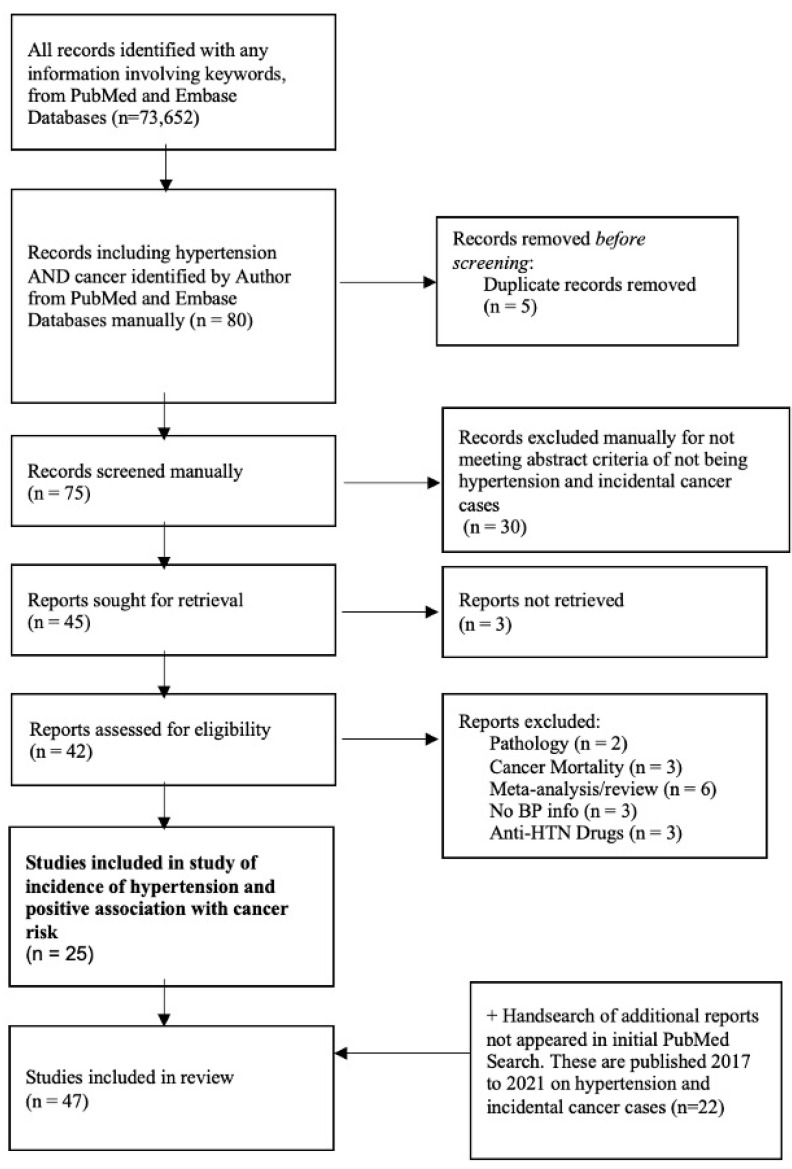
Flowchart of the studies excluded during a manual literature search conducted over five months.

**Figure 2 healthcare-10-01074-f002:**
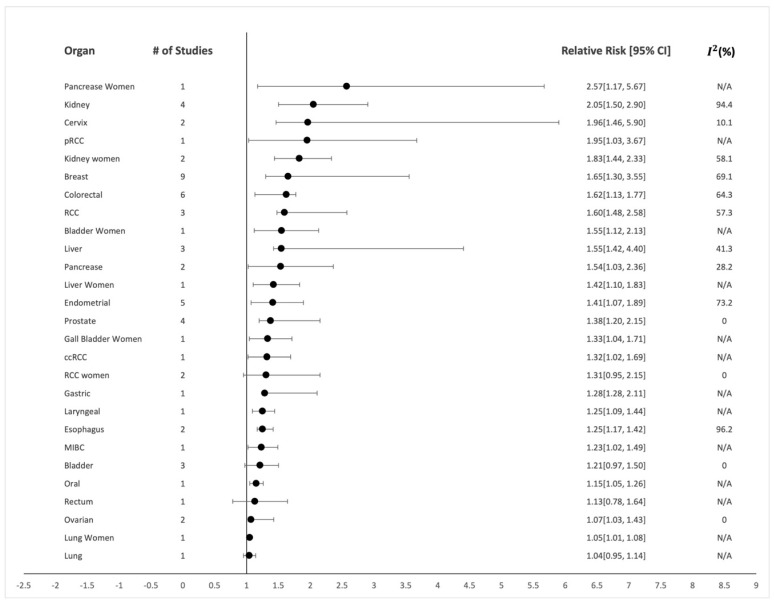
The summary of the relative risks, 95% confidence intervals, and heterogeneity for each individual organ-specific cancer site.

**Figure 3 healthcare-10-01074-f003:**
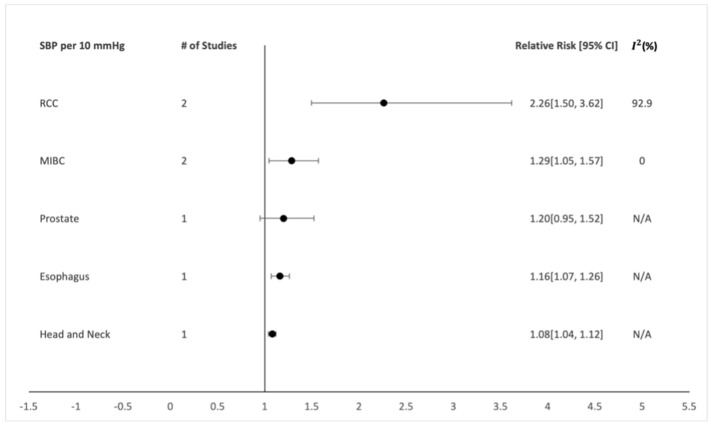
Dose-dependent SBP summary of the relative risks and 95% confidence intervals for each individual organ-specific cancer site.

**Figure 4 healthcare-10-01074-f004:**
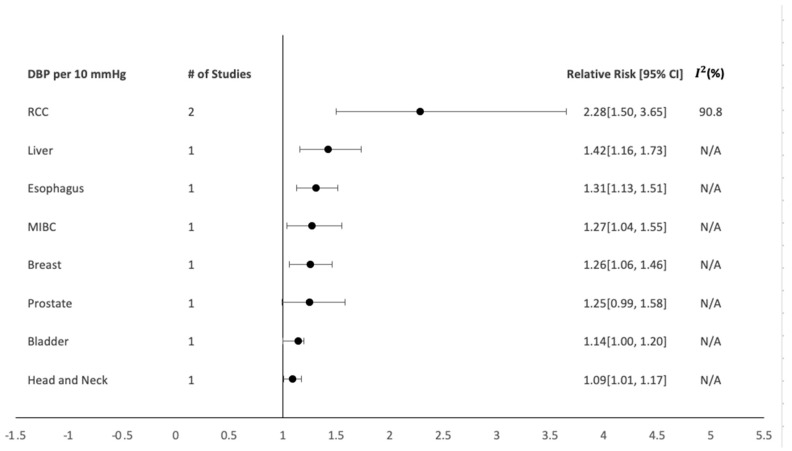
Dose-dependent DBP summary of the relative risks and 95% confidence intervals for each individual organ-specific cancer site.

**Table 1 healthcare-10-01074-t001:** Study characteristics. N/A, not available.

Study	Study Design	Cancer Outcome
Christakoudi, 2020, Europe	Cohort	RCC, Esophagus SCC, Head and Neck
Stocks, 2012, Europe	Cohort	Kidney, Colon, Rectum, Bladder, Pancreas, Liver, Endometrial, Cervix
Lindgren, 2003, Finland	Cohort	Lung
Lindgren, 2007, Finland	Cohort	Breast Cancer
Lindgren, 2005, Finland	Cohort	Kidney
Beebe-Dimmer, 2007, USA	Case-Control	Prostate
Batty, 2003, Europe	Cohort	Pancreas
Behrens, 2016, Denmark	Cohort	Endometrial
Seo, 2020, Korea	Cohort	Esophagus, Oral, Laryngeal
Wallner, 2010, USA	Cohort	Prostate
Martin, 2009, Norway	Cohort	Prostate
Teleka, 2020, Europe	Cohort	Bladder
Teleka, 2018, Europe	Cohort	Bladder
Teleka, 2021, Europe	Cohort	Bladder
Drahos, 2016, USA	Case-Control	Esophagus
Lofterød, 2020, Europe	Cohort	Breast
Sun, 2015, Taiwan	Cohort	Kidney, Endometrial
Shen, 2015, China	Nested Case-Control	RCC
Kim, 2020, Korea	Cohort	Kidney
Yassin, 2019, Palestine	Case-Control	Breast
Staples, 2020, USA	Case-Control	Ovarian
Michels, 2019, USA	Case-Control	Ovarian
Pereira, 2012, Chili	Case-Control	Breast
Chuang, 2015, Taiwan	Nested Case-Control	Breast
Beji, 2007, Turkey	Case-Control	Breast
Colt, 2011, USA	Case-Control	RCC
Jung, 2013, Korea	Case-Control	Breast
Kok, 2018, Taiwan	Cohort	Bladder
Hektoen, 2020, Norway	Cohort	Bladder
Pol, 2020, Netherlands	Cohort	RCC
Ko, 2016, Korea	Cohort	Colon
Li, 2018, China	Case-Control	Gastric
Lee, 2020, Korea	Cohort	Colon
Al-Madani, 2019, Saudi Arabia	Cohort	Cervix
Hirai, 2018, China	N/A	Colon
Samarakoon, 2018, Asia	Case-Control	Colon
Dickerman, 2017, Iceland	Cohort	Prostate
Porto, 2011, Brazil	Case-Control	Breast
Kabat, 2009, USA	Cohort	Breast
Berger, 2016, Denmark	Cohort	Kidney, Liver, Gall Bladder
Choi, 2021, Korea	Cohort	Breast
Jeon, 2020, Korea	Cohort	Lung
Trabert, 2015, USA	Case-Control	Endometrial
Bashamakha, 2019, Yemen	Case-Control	Breast
Flaherty, 2005, USA	Cohort	RCC
Haggstrom, 2013, Europe	Cohort	RCC
Kasmari, 2017, USA	Cohort	Liver

**Table 2 healthcare-10-01074-t002:** Hypertension assessed and defined by each study.

Study	Exposure Assessment	Definition of Hypertension
Christakoudi, 2020, Europe	Measured	BP ≥ 140/90 mmHg
Stocks, 2012, Europe	Measured	BP ≥ 140/90 mmHg
Lindgren, 2003, Finland	Measured	Quartiles
Lindgren, 2007, Finland	Measured	Quartiles
Lindgren, 2005, Finland	Measured	Quartiles
Beebe-Dimmer, 2007, USA	Questionnaire	BP ≥ 140/90 mmHg
Batty, 2003, Europe	Measured	Quartiles
Behrens, 2016, Denmark	Database	N/A
Seo, 2020, Korea	Measured	BP ≥ 140/90 mmHg
Wallner, 2010, USA	Questionnaire	Yes/No
Martin, 2009, Norway	Measured	BP ≥ 130/85 mmHg
Teleka, 2020, Europe	Measured	Quartiles
Teleka, 2018, Europe	Measured	BP ≥ 140/90 mmHg
Teleka, 2021, Europe	Measured	BP ≥ 140/90 mmHg
Drahos, 2016, USA	Database	BP ≥ 140/90 mmHg
Lofterød, 2020, Europe	Measured	BP ≥ 140/90 mmHg
Sun, 2015, Taiwan	Database	Yes/No
Shen, 2015, China	Questionnaire	Yes/No
Kim, 2020, Korea	Database	BP ≥ 130/80 mmHg
Yassin, 2019, Palestine	Questionnaire	N/A
Staples, 2020, USA	Questionnaire	N/A
Michels, 2019, USA	Database	N/A
Pereira, 2012, Chili	Measured	BP ≥ 140/90 mmHg
Chuang, 2015, Taiwan	Database	Yes/No
Beji, 2007, Turkey	Questionnaire	Yes/No
Colt, 2011, USA	Questionnaire	Category of severity
Jung, 2013, Korea	Questionnaire	Yes/No
Kok, 2018, Taiwan	Database	BP ≥ 140/90 mmHg
Hektoen, 2020, Norway	Measured	BP ≥ 140/90 mmHg
Pol, 2020, Netherlands	Questionnaire	Yes/No
Ko, 2016, Korea	Measured	BP ≥ 130/85 mmHg
Li, 2018, China	Measured	BP ≥ 130–140/85–90 mmHg
Lee, 2020, Korea	Measured	Quartiles
Al-Madani, 2019, Saudi Arabia	Questionnaire	Yes/no
Hirai, 2018, China	N/A	N/A
Samarakoon, 2018, Asia	N/A	N/A
Dickerman, 2017, Iceland	Measured	BP ≥ 130/85 mmHg
Porto, 2011, Brazil	Measured	BP ≥ 130/85 mmHg
Kabat, 2009, USA	Measured	BP ≥ 130/85 mmHg
Berger, 2016, Denmark	Questionnaire	Yes/No
Choi, 2021, Korea	Measured	BP ≥ 130/85 mmHg
Jeon, 2020, Korea	Questionnaire	Yes/No
Trabert, 2015, USA	Measured	BP ≥ 130/85 mmHg
Bashamakha, 2019, Yemen	Questionnaire	Yes/No
Flaherty, 2005, USA	Questionnaire	Yes/No
Haggstrom, 2013, Europe	Database	BP ≥ 140/90 mmHg
Kasmari, 2017, USA	Database	Yes/No

## Data Availability

Data will be available per request.

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
