# Peer review of "Association of Hypertension and Organ-Specific Cancer: A Meta-Analysis"

_healthcare, 2022, doi:10.3390/healthcare10061074_

Round 1

Reviewer 1 Report

The study titled “Association of Hypertension and Organ-Specific Cancer: A Meta-Analysis” aims to assess association of hypertension and increased risk of cancer with emphasis on patient specificity and organ specificity. Further, the study also underscored the underlying pathology.

The research question is important in view of rising incidence of hypertension and cancer in the population globally. If the association between hypertension and incidence of cancer is proved then preventive and therapeutic strategies, should look at this association for better preventive measures besides patient management and care.  

    Observations and suggestions:

  • If there is statistics regarding incidence of cancer among hypertensive and non hypertensive, that needs to be included in the background, so as to get a birds eye view of the burden of the problem
  • The need of the meta-analysis can be further strengthened in the form of gaps in the knowledge and how it is going to affect the preventive and therapeutic measures
  • Did the meta-analysis take into consideration the duration of hypertension, severity of hypertension, antihypertensive therapy
  • Does the antihypertensive therapy have any impact on incidence in the form of higher or lower incidence of cancer, organ specific cancer?
  • Does the study take into account the potential confounding factors like age , gender, BMI?
  • Hypertension is partly determined by genetic factors. The same holds true for cancer to an extent. So the authors can highlight on this issue is there any link at the genetic level between hypertension and cancer under the discussion section.
  • Have the authors looked at the confounding factors that were adjusted in the studies. if not, this might introduce heterogeneity and can affect the summary risk estimate. if yes then the authors can summarize the same. 

Author Response

  1. General Comment

The study titled “Association of Hypertension and Organ-Specific Cancer: A Meta-Analysis” aims to assess association of hypertension and increased risk of cancer with emphasis on patient specificity and organ specificity. Further, the study also underscored the underlying pathology.

The research question is important in view of rising incidence of hypertension and cancer in the population globally. If the association between hypertension and incidence of cancer is proved then preventive and therapeutic strategies, should look at this association for better preventive measures besides patient management and care.  

Answer: We thank the reviewer for the supportive comments. We agree that if the association between hypertension and incidence of cancer is proved then preventive and therapeutic strategies, should look at this association for better preventive measures besides patient management and care. Our main aim in this meta-analysis was to show the risk of cancer progression in hypertensive patients and emphasize on importance of preventative strategies.

  1. Major Concerns

  1. If there is statistics regarding incidence of cancer among hypertensive and non hypertensive, that needs to be included in the background, so as to get a birds eye view of the burden of the problem.

Answer: We added following sentences to the introduction:  Paragraph 2, Line 5-8:

“One study observed the odds ratio (OR) with a 95% confidence ratio in individual’s with self-reported hypertension in both men and women and found that the hypertension was independently associated with a 40% increased risk of renal cell carcinoma compared to the refenced non-hypertensive patients (OR, 1.4; CI, 1.1-1.9 and OR, 1.0; CI, reference, respectfully) [26]”

  1. Did the meta-analysis take into consideration the duration of hypertension, severity of hypertension, antihypertensive therapy.

Answer: In this manuscript, our focus has been on showing how high is the risk of organ-specific cancer for hypertension patients. We intentionally excluded studies including patients under non-hypertensive therapy. However, one study specifically compared the risk of breast cancer in patients with/without antihypertensive therapy. It showed that patient under antihypertensive therapy show higher risk.

We have added this study to the discussion: Paragraph 2, Line 2-4; Matrix Metalloproteinases Regulation, Paragraph 5, Line 4-5:

“Specifically, there was a previous study that found that there was an increased risk of developing ER+ breast cancer with those who were long-term ( years) users of antihypertension drugs (RR = 1.21; 95% CI 1.03-1.43) [75].”

Unfortunately, all of our references studied patients who were diagnosed with hypertension but the duration of hypertension before onset of cancer were not discussed. We have added the following sentence to our discussion section as limitation of our analysis: Paragraph 11, Line 4-6

“Another possible limitation is the lack of data on the details of each malignant tumor that developed within these patients and the studies lacked information on how long these tumors developed after the initial hypertension examination.”

  1. Does the antihypertensive therapy have any impact on incidence in the form of higher or lower incidence of cancer, organ specific cancer?

Answer: In this manuscript, our focus has been on showing how high is the risk of organ-specific cancer for hypertension patients. We intentionally excluded studies including patients under non-hypertensive therapy. However, one study specifically compared the risk of breast cancer in patients with/without antihypertensive therapy. It showed that patient under antihypertensive therapy have higher risk.

We have added this study to Discussion, Paragraph 2, Line 2-5:

“Specifically, there was a previous study that found that there was an increased risk of developing ER+ breast cancer with those who were long-term ( years) users of antihypertension drugs (RR = 1.21; 95% CI 1.03-1.43) [75].”

  1. Does the study take into account the potential confounding factors like age , gender, BMI?

Answer: In this manuscript, our main goal is to purely show impact of hypertension on progression of organ-specific cancer. The main challenge of isolating impact of other potential cofounding parameters like age, gender, or BMI is the lack of data purely on that factor. However, we found studies looking at impact of gender on kidney, which is shown in figure 2.

We added the following to the Methods, Paragraph 2, Line 1-2:

“If the study was on metabolic syndrome, which is a combination of factors, such as, BMI, high triglycerides, and hypertension, only hypertension data was examined and extracted.”

We added the following to the introduction Paragraph 3, Line 5-7:

“Since previous studies have demonstrated that hypertension may play a role in cancer, the present study purely focused on untreated hypertensive patients to better understand the pathology between the two and examine possible cross-links between these highly complex diseases to emphasize on the importance of preventative strategies.”

We added following sentences to the Discussion, Matrix Metalloproteinases Regulation, Paragraph 6, Line 1-3

“The present study did not look at contributing factors, such as BMI or genetics in an attempt to gain information purely on hypertension and cancer, so future studies on this research would be beneficial to observe if these contributing factors, along with hypertension, also have an association with developing or not developing incidental cancer.”

  1. Hypertension is partly determined by genetic factors. The same holds true for cancer to an extent. So the authors can highlight on this issue is there any link at the genetic level between hypertension and cancer under the discussion section.

Answer: In our meta-analysis, the included studies have not looked at genetic factors. Patients included only because of being diagnosed with hypertension. We added following sentences to the Discussion, Matrix Metalloproteinases Regulation, paragraph 5, Line 4-17:

“Cancer and hypertension are both extremely complex and may be promoted from various cofounding parameters, such as, BMI, genetics, diabetes, hypertension, high triglycerides, etc. A previous study that did look at other factors associated with metabolic syndrome, found that the strongest association with prostate cancer was a combination of obesity and hypertension, but when they studied these factors individually, hypertension independently was associated the most strongly with prostate cancer risk [48]. Another study evaluated whether the combination of comorbid conditions was associated with epithelial ovarian cancer (EOC), and they found that women who had untreated hypertension for less than 10 years were at the strongest risk. Importantly, they also found that women diagnosed with all three, hypertension, hyperlipidemia, and diabetes, had a decreased risk of EOC [54]. Other studies have also demonstrated that out of all the metabolic factors associated with cancer, hypertension plays a very strong role [17,29,35-36] in cancer initiation. Since previous studies have demonstrated that hypertension may be associated with cancer at a higher risk than other metabolic factors, the present study purely focused on untreated hypertensive patients to better understand the pathology between the two and examine possible cross-links between these highly complex diseases to emphasize on the importance of preventative strategies. Our study serves as an important first step in understanding the association between hypertension and different organ-specific cancer incidents. Both cancer and hypertension, as mentioned, are both partly determined by genetic factors. However, identifying any common genetic link between hypertension and cancer is beyond scope of this study. “

  1. Have the authors looked at the confounding factors that were adjusted in the studies. if not, this might introduce heterogeneity and can affect the summary risk estimate. if yes then the authors can summarize the same. 

Answer: In this manuscript, our main goal is to purely show impact of hypertension on progression of organ-specific cancer. In our meta-analysis, we only included studies looking at the association of organ-specific cancer incidences in hypertensive patients.

We added following sentences to the Discussion, Matrix Metalloproteinases Regulation, paragraph 5, Line 5-15:

“A previous study that did look at other factors associated with metabolic syndrome, found that the strongest association with prostate cancer was a combination of obesity and hypertension, but when they studied these factors individually, hypertension independently was associated the most strongly with prostate cancer risk [48]. Another study evaluated whether the combination of comorbid conditions was associated with epithelial ovarian cancer (EOC), and they found that women who had untreated hypertension for less than 10 years were at the strongest risk. Importantly, they also found that women diagnosed with all three, hypertension, hyperlipidemia, and diabetes, had a decreased risk of EOC [54]. Other studies have also demonstrated that out of all the metabolic factors associated with cancer, hypertension plays a very strong role [17,29,35-36] in cancer initiation. Since previous studies have demonstrated that hypertension may be associated with cancer at a higher risk than other metabolic factors, the present study purely focused on untreated hypertensive patients to better understand the pathology between the two and examine possible cross-links between these highly complex diseases to emphasize on the importance of preventative strategies.”

We added following sentences to the Discussion, Matrix Metalloproteinases Regulation, paragraph 7

“In conclusion, since hypertension and cancer are both extremely complex, there are several possible parameters that could also play a role in cancer initiation and progression. The present study focused on hypertension because of its pathological complexity and lack of information on the possible cross-link between hypertension and cancer. The present study was able to affectively show that one of these factors, hypertension, may contribute to carcinogenesis and that this association between hypertension and cancer is observed between different organ-specific cancer sites. Specifically, individuals with hypertension were at a high risk of kidney, breast, colorectal, endometrial, and bladder cancers. This study was also able to evaluate the possible mechanisms behind this and enhance the understanding of this phenomenon and give detailed information on relevant information that may have the potential to improve preventable and therapeutic treatments for these patients.”

Reviewer 2 Report

We don't think the conclusion of this article is reliable. In fact, the relationship between hypertension and cancers is affected by many confounding factors, and the literature included in each tumor is also limited.

Author Response

Answer: We agree with reviewer that there are several other parameters which cause cancer. In this manuscript, our main goal is to purely show impact of hypertension on progression of organ-specific cancer. In our meta-analysis, we only included studies looking at the association of organ-specific cancer incidences in hypertensive patients.

We added following sentences to the Discussion, Matrix Metalloproteinases Regulation, paragraph 5, Line 5-15:

“A previous study that did look at other factors associated with metabolic syndrome, found that the strongest association with prostate cancer was a combination of obesity and hypertension, but when they studied these factors individually, hypertension independently was associated the most strongly with prostate cancer risk [48]. Another study evaluated whether the combination of comorbid conditions was associated with epithelial ovarian cancer (EOC), and they found that women who had untreated hypertension for less than 10 years were at the strongest risk. Importantly, they also found that women diagnosed with all three, hypertension, hyperlipidemia, and diabetes, had a decreased risk of EOC [54]. Other studies have also demonstrated that out of all the metabolic factors associated with cancer, hypertension plays a very strong role [17,29,35-36] in cancer initiation. Since previous studies have demonstrated that hypertension may be associated with cancer at a higher risk than other metabolic factors, the present study purely focused on untreated hypertensive patients to better understand the pathology between the two and examine possible cross-links between these highly complex diseases to emphasize on the importance of preventative strategies.”

We added following sentences to the Discussion, Matrix Metalloproteinases Regulation, paragraph 6, Line 4-6

“Another possible limitation is the lack of data on the details of each malignant tumor that developed within these patients and the studies lacked information on how long these tumors developed after the initial hypertension examination. “

We added following sentences to the conclusion:

“In conclusion, since hypertension and cancer are both extremely complex, there are several possible parameters that could also play a role in cancer initiation and progression. The present study focused on hypertension because of its pathological complexity and lack of information on the possible cross-link between hypertension and cancer. The present study was able to affectively show that one of these factors, hypertension, may contribute to carcinogenesis and that this association between hypertension and cancer is observed between different organ-specific cancer sites. Specifically, individuals with hypertension were at a high risk of kidney, breast, colorectal, endometrial, and bladder cancers. This study was also able to evaluate the possible mechanisms behind this and enhance the understanding of this phenomenon and give detailed information on relevant information that may have the potential to improve preventable and therapeutic treatments for these patients.”

Reviewer 3 Report

This meta-analytic paper sheds light on an unusual aspect of oncogenesis and adds some statistically interesting details to support the mutual aspect of carcinogenesis and hypertension.

I wonder if the authors could demonstrate that increased BMI could also be a risk factor for each disorders.

I also wonder if familial accumulation of different types of cancers have been shown in any of the involved clinical studies.

Since carcinogenesis is obviously a very complicated process if I were the authors my conclusion would have been very moderate and delicate and would handle the relative risk provided by hypertension very "softly and gently'".

Author Response

  1. General Comment

This meta-analytic paper sheds light on an unusual aspect of oncogenesis and adds some statistically interesting details to support the mutual aspect of carcinogenesis and hypertension.

Answer: We thank the reviewer for the supportive comments. We agree that if the association between hypertension and incidence of cancer is proved then preventive and therapeutic strategies, should look at this association for better preventive measures besides patient management and care. Our main aim in this meta-analysis was to show the risk of cancer progression in hypertensive patients and emphasize on importance of preventative strategies.

  1. Major Concerns

  1. I wonder if the authors could demonstrate that increased BMI could also be a risk factor for each disorders.

Answer: In this manuscript, our main goal is to purely show impact of hypertension on progression of organ-specific cancer. However, impact of gender was discussed in all cancer types. In our meta-analysis, we only included studies looking at the association of organ-specific cancer incidences in hypertensive patients.

We added following sentences to the Discussion, Matrix Metalloproteinases Regulation, paragraph 5, Line 5-15:

“A previous study that did look at other factors associated with metabolic syndrome, found that the strongest association with prostate cancer was a combination of obesity and hypertension, but when they studied these factors individually, hypertension independently was associated the most strongly with prostate cancer risk [48]. Another study evaluated whether the combination of comorbid conditions was associated with epithelial ovarian cancer (EOC), and they found that women who had untreated hypertension for less than 10 years were at the strongest risk. Importantly, they also found that women diagnosed with all three, hypertension, hyperlipidemia, and diabetes, had a decreased risk of EOC [54]. Other studies have also demonstrated that out of all the metabolic factors associated with cancer, hypertension plays a very strong role [17,29,35-36] in cancer initiation. Since previous studies have demonstrated that hypertension may be associated with cancer at a higher risk than other metabolic factors, the present study purely focused on untreated hypertensive patients to better understand the pathology between the two and examine possible cross-links between these highly complex diseases to emphasize on the importance of preventative strategies.”

We added following sentences to the Discussion, Matrix Metalloproteinases Regulation, Paragraph 5, Line 1-3

“The present study did not look at contributing factors, such as BMI or genetics to gain information purely on hypertension and cancer, so future studies on this research would be beneficial to observe if these contributing factors, along with hypertension, also have an association with developing or not developing incidental cancer.”

  1. I also wonder if familial accumulation of different types of cancers have been shown in any of the involved clinical studies.

Answer: In our meta-analysis, the included studies have not looked at genetic factors. Patients included only because of being diagnosed with hypertension. We added following sentences to the Discussion, Matrix Metalloproteinases Regulation, paragraph Discussion, Matrix Metalloproteinases Regulation, paragraph 5, Line 4-17:

“Cancer and hypertension are both extremely complex and may be promoted from various cofounding parameters, such as, BMI, genetics, diabetes, hypertension, high triglycerides, etc. A previous study that did look at other factors associated with metabolic syndrome, found that the strongest association with prostate cancer was a combination of obesity and hypertension, but when they studied these factors individually, hypertension independently was associated the most strongly with prostate cancer risk [48]. Another study evaluated whether the combination of comorbid conditions was associated with epithelial ovarian cancer (EOC), and they found that women who had untreated hypertension for less than 10 years were at the strongest risk. Importantly, they also found that women diagnosed with all three, hypertension, hyperlipidemia, and diabetes, had a decreased risk of EOC [54]. Other studies have also demonstrated that out of all the metabolic factors associated with cancer, hypertension plays a very strong role [17,29,35-36] in cancer initiation. Since previous studies have demonstrated that hypertension may be associated with cancer at a higher risk than other metabolic factors, the present study purely focused on untreated hypertensive patients to better understand the pathology between the two and examine possible cross-links between these highly complex diseases to emphasize on the importance of preventative strategies. Our study serves as an important first step in understanding the association between hypertension and different organ-specific cancer incidents. Both cancer and hypertension, as mentioned, are both partly determined by genetic factors. However, identifying any common genetic link between hypertension and cancer is beyond scope of this study. “

We added following sentences to the Discussion, Matrix Metalloproteinases Regulation, Paragraph 6, Line 1-3

“The present study did not look at contributing factors, such as BMI or genetics in an attempt to gain information purely on hypertension and cancer, so future studies on this research would be beneficial to observe if these contributing factors, along with hypertension, also have an association with developing or not developing incidental cancer.”

  1. Since carcinogenesis is obviously a very complicated process if I were the authors my conclusion would have been very moderate and delicate and would handle the relative risk provided by hypertension very "softly and gently'".

Answer: We thank the reviewer for his suggestion. We revised the conclusion:

“In conclusion, since hypertension and cancer are both extremely complex, there are several possible parameters that could also play a role in cancer initiation and progression. The present study focused on hypertension because of its pathological complexity and lack of information on the possible cross-link between hypertension and cancer. The present study was also able to demonstrate that one of these factors, hypertension, may contribute to carcinogenesis and that this association between hypertension and cancer is possibly observed between different organ-specific cancer sites. Specifically, the present study observed that individuals with hypertension were at a high risk of kidney, breast, colorectal, endometrial, and bladder cancers. It is important to note that the high complexity of hypertension and cancer makes it difficult to give a definite remark as to why and how this is happening, but this study was also able to give detailed evaluations of the possible mechanisms of this observed cross-linkage and attempted to enhance the understanding of this phenomenon and give detailed information on relevant information that may have the potential to improve preventable and therapeutic treatments for these patients.”

Reviewer 4 Report

The current review investigates the prevalence of hypertension and cancer comorbidity. The authors assess which organ-specific cancers are associated with hypertension.  It is well written and includes comprehensive information. The abstract describes the study and the findings. The content arrangement of the manuscript is reasonable. The search criteria were clearly defined. The authors provide a diagram of the studies excluded and included. The summary relative risks of hypertension for each organ-specific cancer site were also shown.  They discussed the most common pathways involved in hypertension-causes cancer as could be used in designing cancer prevention strategies for these patients.

Author Response

Answer: We thank the reviewer for the supportive comments.

Round 2

Reviewer 1 Report

The queries have been addressed and found satisfactory

Reviewer 2 Report

Accept in present form